# 90-Day Patient-Centered Outcomes after Totally Endoscopic Cardiac Surgery: A Prospective Cohort Study

**DOI:** 10.3390/jcm11092674

**Published:** 2022-05-09

**Authors:** Jade Claessens, Alaaddin Yilmaz, Toon Mostien, Silke Van Genechten, Marithé Claes, Loren Packlé, Maud Pierson, Jeroen Vandenbrande, Abdullah Kaya, Björn Stessel

**Affiliations:** 1Faculty of Medicine and Life Sciences, UHasselt—Hasselt University, Agoralaan, 3590 Diepenbeek, Belgium; bjorn.stessel@jessazh.be; 2Department of Cardiothoracic Surgery, Jessa Hospital, Stadsomvaart 11, 3500 Hasselt, Belgium; alaaddin.yilmaz@jessazh.be (A.Y.); silke_v@msn.com (S.V.G.); loren.packle@jessazh.be (L.P.); abdullah.kaya@jessazh.be (A.K.); 3Department of Anesthesiology, Jessa Hospital, Stadsomvaart 11, 3500 Hasselt, Belgium; toon.mostien@jessazh.be (T.M.); marithe.claes@uhasselt.be (M.C.); piersonmaud@gmail.com (M.P.); jeroen.vandenbrande@jessazh.be (J.V.)

**Keywords:** quality of life, clinical outcomes, totally endoscopic cardiac surgery

## Abstract

Over the past years, minimally invasive procedures have been developed to reduce surgical trauma after cardiac surgery. The value of patient-centered outcomes, including the quality of recovery after hospital discharge, is increasingly recognized. Identifying meaningful changes in postoperative function that might have a negative impact on patients without noticeable complications can provide a more comprehensive understanding of the impact on the patient’s life. In total, 209 patients were included in this trial. Of these, 193 patients underwent totally endoscopic cardiac surgery, 8 underwent cardiac surgery through a sternotomy, and 8 underwent transcatheter aortic valve implantation. Patients who previously underwent cardiac surgery were excluded. Quality of life was determined through the Short Form 36 and European Quality of Life-5 Dimensions questionnaires before the surgery and 14, 30, and 90 days afterward. In patients who underwent totally endoscopic cardiac surgery, the quality of life improved over the three time periods. The different domains of the questionnaire evolved in a positive manner. However, 14 days postoperatively, a decline in quality of life was noted, followed by a return to baseline at 30 days and an increase at 90 days. In conclusion, totally endoscopic cardiac surgery improves the quality of life 90 days after surgery.

## 1. Introduction

The conventional access route to conduct cardiac surgery is a median sternotomy. This technique, however, is associated with a longer hospital stay and high healthcare costs [1]. Furthermore, it includes a 17–56% risk of post-sternotomy pain syndrome [2] and a negative patient experience due to poor cosmetic results [3]. Therefore, research mainly focuses on reducing cardiac surgical trauma by using less invasive access to the heart.

In 2020, Yilmaz et al. introduced a new totally endoscopic CABG (endo-CABG) technique using only endoscopic instruments [4]. This technique seems to be associated with a lower 30-day mortality rate (1.8%) compared to conventional CABG (2.5–2.8%) [4,5]. Furthermore, endo-CABG is less expensive and less time-consuming than robotic CABG procedures [4]. In the field of aortic valve replacement (AVR), several types of less invasive procedures were introduced, such as right anterolateral mini-thoracotomy (RALT) and right anterior mini-thoracotomy (RAMT) [6,7,8,9]. The first totally endoscopic AVR, using five trocars, was performed successfully by Vola et al. in 2014 [10]. Additionally, transcatheter aortic valve implantation (TAVI) through the femoral artery has become a popular alternative and may offer a good option for patients who are not eligible for surgery [11]. In mitral valve surgery, endoscopic techniques were already described in 1997 [12]. However, mitral valve surgery through video-assisted thoracoscopic surgery (MVATS) did not gain widespread application due to the technical challenge of this technique and the need for surgical experience [13].

Patient-centered outcomes, including health-related quality of life (HRQL), have become important endpoints in medical care. These outcome measures are reports of the patient’s health status and the influence on their daily lives without a professional interpretation [14]. HRQL questionnaires, such as the Short Form 36 (SF-36) and the EuroQoL-5 Dimension (EQ-5D), enable a more comprehensive understanding of the impact of surgery on the patient’s health. Even though surgery is successful, HRQL can be discouraging for some patients [15]. The overall quality of care in patients undergoing totally endoscopic cardiac surgery (TECS) may also be improved by uncovering potential limitations in the care process [14]. HRQL questionnaires may fulfill the requirements as valuable indicators of surgical quality of recovery (QoR) [16].

The aforementioned techniques are not yet widely used, and as a result, prospective data concerning patient-centered and clinical outcomes is relatively scarce. Hence, this study aims to prospectively assess both patient-centered and clinical outcomes after these minimally invasive surgical procedures.

## 2. Materials and Methods

### 2.1. Study Design

This prospective longitudinal cohort study was approved by the local ethics committee of the Jessa Hospital Belgium (registration number B243201836445) and registered on clinicaltrials.gov (NCT03902717). This paper is conducted in accordance with the Declaration of Helsinki. Written consent was obtained from all participants before starting.

All patients undergoing TECS, TAVI or conventional open CABG between November 2019 and October 2020 were eligible to participate. TECS procedures include totally endoscopic AVR (Yil-AVR), endo-CABG, MVATS, and a combination of endoscopic procedures. Exclusion criteria were age <18 years, participation in another trial, previous cardiac surgery, conversion to sternotomy, inability to understand the study or insufficient understanding of the Dutch language.

The primary outcome of this study is HRQL after different types of TECS procedures. HRQL is assessed by the SF-36 and EQ-5D questionnaires. Both are validated self-reporting questionnaires and can be completed in approximately 10 min. These questionnaires were taken at baseline, the day before surgery, and subsequently on postoperative days 14, 30, and 90. To investigate the entire recovery process and create a 90-day QoR profile, 14- and 30-day, and 90-day measurements were taken.

### 2.2. Surgical Techniques

The description of the endo-CABG technique is published in J. Cardiology [4]. This procedure is performed through three endoscopic ports (5 mm) in the 2nd, 3rd, and 4th intercostal space and a 2–3 cm utility port. Additionally, in totally endoscopic AVR, aortic access is gained by a 2 cm working port in the 2nd intercostal space and three 5mm trocars in the 2nd and 3rd intercostal spaces, using zero-degree optics. For the MVATS procedure, three 5–15 mm incisions are made. In all procedures, peripheral cardiopulmonary bypass is initiated, followed by transthoracic aortic cross-clamping and antegrade administration of a single shot cold mixed-blood cardioplegia.

### 2.3. Quality of Life

The SF-36 is widely used to measure HRQL [17]. This validated questionnaire covers the physical and mental health of a patient based on eight categories: physical functioning, role limitations due to physical health, pain, general health, role limitations due to emotional problems, energy/fatigue, emotional wellbeing, and social functioning. The domains allow for the calculation of a physical and mental component score (PCS and MCS). In this way, the eight SF-36 scales are standardized using means and standard deviations from a reference population of ischemic heart disease in Belgium.

In another standardized method, the EQ-5D-5L questionnaire, patients rate “their health today” based on five dimensions: mobility, self-care, daily activities, pain/discomfort, and anxiety. Of these five dimensions, an index value can be calculated based on a crosswalk value set of a specific population (general population of the Netherlands). The EQ visual analog scale (VAS) scores the patient’s “health today” on a scale from 0 to 100 [18].

The difference between baseline HRQL and HRQL at 14, 30 and 90 days after surgery was calculated to measure the degree of recovery. We applied the following definition of QoR, based on the SF-36-score: QoR was predefined into recovered and improved. Recovered was defined as the absence of a significant difference between the total median postoperative SF-36 score and baseline SF-36 score. Improved was described as a substantial improvement in total median postoperative SF-36 score compared with baseline [19,20]. An SF-36 score between one point lower and four points higher than the baseline SF-36 score was considered recovered. An SF-36 score five points higher than baseline was considered improved. Poor QoR was defined as “failed to recover” at 30 days and “failed to improve” at 90 days [20,21].

### 2.4. Statistical Analysis

The normality of the data was assessed using the Shapiro–Wilk test. Continuous variables were presented as the median and interquartile range (IQR). Categorical variables were described as numbers and corresponding percentages. The different types of surgical procedures were compared using a Kruskal–Wallis test for non-parametric data, while a Chi-square test was used for categorical variables. The Friedman and Wilcoxon signed-rank tests analyzed changes between the baseline and postoperative HRQL. A linear mixed model was used to compare the HRQL between two types of surgery. Univariate analysis was performed to determine significant predictors of a poor QoR, i.e., 30-day recovery and 90-day improvement. Candidate variables with a *p*-value of <0.10 were considered for the multivariate regression analysis. An optimal regression model was formed using backward elimination. A *p*-value smaller than or equal to 0.05 was considered significant. All statistical analyses were performed using the R Core Team (2021). R: A language and environment for statistical computing. R Foundation for Statistical Computing, Vienna, Austria.

## 3. Results

A STROBE flow chart of patient selection and exclusion is presented in Figure 1.

### 3.1. Demographics

Overall, the baseline characteristics were similar in all subpopulations. However, the TAVI group was significantly older than the Yil-AVR group (85.5 (77.75–87.75) versus 73 (65–76), *p* < 0.001). Additionally, the mean European System for Cardiac Operative Risk Evaluation (Euroscore) II was significantly higher in the open CABG group compared with Endo-CABG (3.54 (1.90–4.55) versus 1.29 (0.93–2.17), *p* = 0.021). Demographics are shown in Table 1.

### 3.2. Quality of Life

The Physical Component Score (PCS) of the SF-36 in the overall TECS population significantly changed after surgery (*p* < 0.001, Figure 2A). At 14 days, the PCS was significantly lower compared to baseline (53.98 (43.91–60.67) versus 50.27 (43.48–56.13), respectively, *p* = 0.004). At 30 days, the PCS was returned to baseline values (*p* = 0.811), and at 90 days, it was significantly improved (61.64 (55.05–65.64) at 90 days, *p* < 0.001). In the endo-CABG (Appendix A), Yil-AVR (Appendix A), MVATS, and combination subdivisions, the same significant improvements were observed after 90 days (*p* < 0.001, *p* < 0.001, *p* = 0.028, *p* = 0.001, respectively). In contrast to the PCS, the Mental Component Score (MCS) did not change significantly after TECS (Figure 2B).

Compared to open CABG, the PCS was significantly better in the endo-CABG group at different points in time (*p* = 0.012, Figure 3A). Same wise, the MCS was not significantly different (*p* = 0.225, Figure 3B). When comparing Yil-AVR and TAVI, no significant difference in the PCS and MCS was seen (*p* = 0.878 and *p* = 0.815, respectively, Figure 3C,D). Additionally, isolated TECS procedures compared to combination TECS showed no significantly different (*p*= 0.305 and *p* = 0.436, respectively, for the PCS and MCS, Figure 3E,F).

The index score of the EQ-5D questionnaire significantly improved over time in the overall TECS population (*p* < 0.001) (Figure 2C). A significant decline was observed after 14 days (*p* < 0.001), followed by a return to baseline values at 30 days and a significant increase at 90 days (*p* = 0.015). A similar evolution was found in the endo-CABG subpopulation. After Yil-AVR, a decline at 14 days and a subsequent return to baseline at 30 and 90 days was observed. Additionally, the EQ-VAS and index score results were similar in the overall TECS (Figure 2D) and endo-CABG populations.

*Physical health*—Overall, the physical functioning significantly increased over time (*p* < 0.001) (Figure 2E). At 14 days, a significant decline was observed (*p* < 0.001), followed by a return to baseline at 30 days, and increased significantly at 90 days (*p* < 0.001). A similar pattern was observed in the endo-CABG group. Contrary to other TECS procedures, Yil-AVR patients showed a significant improvement in physical functioning over time (*p* < 0.001) without a decline at 14 days. On the other hand, patients reported no significant changes in role limitations due to physical health over time (Figure 2F). The pain score of the entire TECS group is presented in Figure 2G. In total, 37.11% reported any form of pain after 90 days. Only 0.6% reported severe pain. Additionally, A significant improvement in general health was observed in the entire TECS group (*p* < 0.001, Figure 2H), after endo-CABG (*p* < 0.001), and after Yil-AVR (*p* = 0.007).

*Mental health*—Patients reported no significant changes in role limitations due to emotional health over time after TECS (Figure 2I). The scores remained high over the four time points. Concerning their emotional wellbeing, TECS patients reported an overall improvement (*p* = 0.004) (Figure 2J). Moreover, the energy or fatigue scores over time are shown in Figure 2K. Compared to baseline, a significant decrease in social functioning was seen after 14 days (*p* < 0.001), improving after 30 and 90 days. Still, it remained significantly lower than baseline (*p* = 0.013 and *p* < 0.001, respectively) (Figure 2L).

### 3.3. Perioperative Clinical Outcomes

All clinical outcomes are presented in Table 2. All TECS procedures were performed successfully without conversion to full sternotomy and no in-hospital mortality. Fourteen patients needed prolonged ventilation (more than 24 h). ICU LOS was significantly higher after Yil-AVR than after TAVI (*p* = 0.041). Excessive bleeding necessitating revision (<24 h) was performed in ten TECS cases (5.21%). Another four patients needed a reinspection within one week after surgery. The all-cause 30-day mortality rate was 2.59%.

### 3.4. Predictors of QoR

After 30 days, 100 (51.81%) patients were considered recovered. After 90 days, 88 (45.60%) patients were deemed to be improved compared to baseline. All variables tested in the univariate analyses are described in Appendix A. Age, CPB time, AHT and smoking were negatively correlated with a 30-day recovery (*p* < 0.05). Euroscore II, ICU LOS, hospital LOS, educational level, combinations of surgeries, and clamping time were also correlated with a 30-day recovery (*p* < 0.1) and were also included in the multivariate logistic regression model (Table 3). Multivariate analysis showed a history of AHT and a longer ICU LOS to be independent predictors of a poor QoR 30 days postoperatively (Table 3). Univariate analysis showed that BMI, hospital LOS and clamping time were correlated with a 90-day improvement. Multivariate analysis showed a longer hospital LOS and shorter clamping time as independent predictors of a poor QoR 90 days postoperatively (Table 3).

## 4. Discussion

### 4.1. Overall HRQL

The present study indicates that the subjectively perceived overall HRQL, as well as physical health and general health level, is decreased 14 days after TECS. Subsequently, these levels return to baseline at 30 days and significantly improve 90 days after TECS. Similar results were observed in the procedure-specific groups. Furthermore, although the baseline PCS of the SF-36 was similar in the open CABG group and endo-CABG group, PCS was higher after endo-CABG at all three postoperative data points. Finally, no differences could be observed in PCS between combination TECS and no combination TECS nor between Yil-AVR and TAVI. These results suggest that TECS positively impacts the subjective perceived physical and general health three months after surgery.

Our results on overall HRQL and general health after TECS align with Bonaros et al., who also observed a general health level equal to the baseline level at 30 days after TECAB and a significant increase 90 days after TECAB [22]. The finding of an initial drop of overall HRQL after surgery, followed by a return to the baseline level, and subsequent significant improvement are also consistent with other HRQL studies of minimally invasive mitral valve surgery [23,24]. This phenomenon can, of course, be explained by surgical trauma. Patients need time to recover from the surgery itself.

### 4.2. Physical Health

Physical functioning levels are higher after TECS than open cardiac surgery during the entire 90-day follow-up period. Despite a higher Euroscore II, prolonged CPB and clamping time, and surgical difficulty, combination TECS seems not to be associated with lower postoperative physical functioning levels. Even though Yil-AVR, compared to TAVI, is a procedure using invasive extracorporeal circulation, it did not translate into less postoperative physical health levels. However, the TAVI population was a smaller cohort and was significantly older than the Yil-AVR.

Moreover, in our series, 37.11% of patients reported a certain degree of pain at 90 days in this study, but only 0.6% reported severe pain. This physical health is substantially less than traditional sternotomy, with severe pain in 17–56% of patients [2].

Parallel with our results on physical functioning, patients also experience progressively increasing energy levels at 30 and 90 days postoperatively, after an initial drop in energy score two weeks after TECS. This result echoes those of a previous cohort study: in TECAB patients, the energy levels at 30 days were equal to baseline, while at 90 days, they significantly increased [22].

### 4.3. Mental Health

Our mental health scores suggest that TECS does not significantly affect patients’ mental health. Inline, patients do not seem to experience significant role limitations due to their emotional health before or after TECS. These results may indicate that neither their underlying health condition nor their surgery predominantly affects this domain. In contrast, role limitations due to emotional health were improved after mitral valve regurgitation through port access [24].

### 4.4. Predictors of Recovery

Another important goal of this study was to identify predictors of a poor QoR at 30 and 90 days after TECS. Our data showed that a history of AHT and longer ICU LOS were independent predictors of a poor QoR 30 days postoperatively. At 90 days after TECS, shorter clamping times and longer hospital LOS were independent predictors of a poor QoR. Literature, which evaluates predictors of a poor QoR after TECS, is scarce. The predictive value of a history of AHT and a longer ICU/hospital LOS for a poor QoR seems logical and does not need further clarification. The association between longer clamping times and a better 90-day improvement seems counter-intuitive. Patients undergoing longer clamping times may have a lower baseline HRQL and, therefore, reach a more easily improved status at 90 days. Albeit, taking the odds ratios into account, only a history of AHT seems to have significant predictive power.

### 4.5. Clinical Outcomes

Cross-clamping and CPB times were similar to or shorter than previous studies [4,22,23]. The neurological outcomes (cerebrovascular accident (CVA): 1.04%; transient ischemic attack (TIA):1.04%) were within the normal range after cardiac surgery [25]. After conventional CABG, the median hospital LOS in Belgium is 11 days [26]. A previous study on endo-CABG reported a hospital LOS of 8 days [4]. In this study, hospital LOS after endo-CABG was only 5 days. Hospital LOS after Yil-AVR was also 5 days compared to previous studies reporting a hospital LOS of 6–10 days after AVR through mini-thoracotomy [27,28]. A lower 30-day mortality rate (0%) of endo-CABG patients was observed in this study compared to conventional CABG (1–3%), MIDCAB (0.8–1.9%), and the previous endo-CABG study (1.8%) [29,30].

### 4.6. Limitations

This observational cohort study includes some limitations. Firstly, the sample size of the control group was too small to draw firm conclusions. Secondly, the Hawthorne effect should be borne in mind since some patients may have reported a more positive HRQL due to study participation effects [31]. Thirdly, the SF-36 has been criticized for significant floor and ceiling effects [32]. Finally, the COVID-19 pandemic may have influenced the results, especially the patients’ mental health.

## 5. Conclusions

In conclusion, all types of TECS resulted in an improved quality of life 90 days after surgery. Predictors of a poor QoR at 30 days were a history of arterial hypertension and longer ICU LOS. Predictors of a poor QoR at 90 days were shorter clamping time and longer hospital LOS.

## Figures and Tables

**Figure 1 jcm-11-02674-f001:**
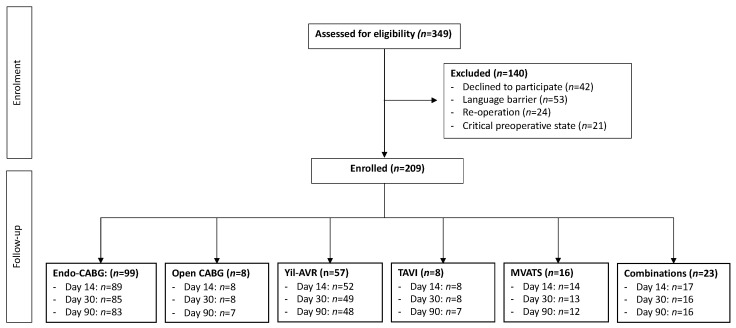
Flowchart of patient recruitment and follow-up. Endo-CABG: endoscopic coronary artery bypass graft; Yil-AVR: endoscopic aortic valve replacement; TAVI: transcatheter aortic valve implantation; MVATS: mitral valve surgery through video-assisted thoracoscopic surgery.

**Figure 2 jcm-11-02674-f002:**
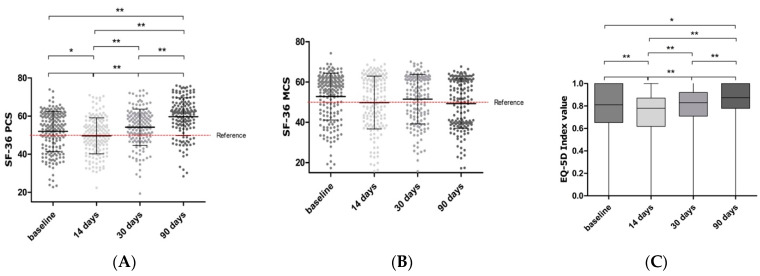
Different domains of the Short Form 36 (SF-36) and Euro Quality of Life (EQ-5D) questionnaires after totally endoscopic cardiac surgery. A physical component score (PCS; (**A**)) and metal component score (MCS; (**B**)) was calculated from the different domains of the SF-36. These included physical functioning (**E**), role limitations physical (**F**), pain (**G**), general health (**H**), role limitations emotional (**I**), emotional wellbeing (**J**), energy/fatigue (**K**) and social functioning (**L**). The EQ-5D is represented through the index value (**C**) and the Visual Analogue Scale (VAS; (**D**)). Data are shown as median and interquartile ranges. The reference line represents the mean of a reference population (ischemic heart disease in Belgium). Significance is indicated as * *p* < 0.05; ** *p* < 0.001.

**Figure 3 jcm-11-02674-f003:**
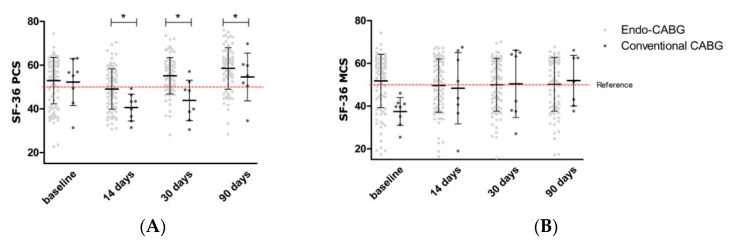
Comparison of the physical and mental component score (PCS and MCS) between Endo-CABG and open CABG (**A**,**B**); Yil-AVR and TAVI (**C**,**D**); and combi and no combi (**E**,**F**). Data are shown as median and interquartile ranges. The reference line represents the mean of a reference population (ischemic heart disease in Belgium). Significance is indicated as * *p* < 0.05.

**Table 1 jcm-11-02674-t001:** Demographics and medical history. Data is represented as *n* (%) and median (IQR).

	TECS (*n* = 193)	Endo-CABG (*n* = 99)	Open CABG (*n* = 8)	*p*-Value	Yil-AVR (*n* = 57)	TAVI (*n* = 8)	*p*-Value	MVATS (*n* = 16)	Combination (*n* = 23)
Age (years)	70 (62–77)	67 (61.5–73.5)	70 (68–75.5)	0.441	73 (65–76)	85.5 (77.75–87.75)	<0.001	72.5 (64.75–80.25)	75 (69–78)
BMI (kg/m^2^)	26.87 (25–29.75)	26.73 (25.09–29.61)	27.84 (22.66–29.43)	0.859	27.04 (23.99–30.85)	26.04 (24.47–26.91)	0.259	26. 11(22.86–28.1)	27.59 (26.02–30.93)
Euroscore II (%)	1.55 (1.04–2.62)	1.29 (0.93–2.17)	3.54 (1.90–4.55)	0.021	1.37 (1.04–2.31)	-	-	2.25 (1.93–3.35)	2.69 (2.01–6.04)
Gender (male)	146 (75.65)	86 (86.87)	6 (75)	0.352	33 (57.89)	5 (62.50)	0.805	13 (81.25)	15 (65.22)
*Smoking*				0.277			0.146		
Active	42 (21.76)	24 (24.24)	4 (50)		11 (19.3)	0 (0)		5 (31.25)	3 (13.04)
Stopped	38 (19.69)	22 (22.22)	1 (12.50)		8 (14.04)	3 (37.50)		1 (6.25)	7 (30.43)
*DiM*				0.840			0.480		
Type I	4 (2.07)	3 (3.03)	0 (0)		1 (1.75)	0 (0)		0 (0)	0 (0)
Type II	44 (22.8)	29 (29.29)	2 (25)		6 (10.53)	2 (25)		3 (18.75)	7 (30.43)
AHT	130 (67.36)	70 (70.7)	7 (87.5)	0.309	35 (61.40)	5 (62.50)	0.952	7 (43.75)	20 (86.96)
*Profession*				0.799			0.068		
Independent contractor	11 (5.7)	7 (7.07)	0 (0)		4 (7.02)	0 (0)		0 (0)	0 (0)
Employed	18 (9.33)	9 (9.09)	1 (12.5)		6 (10.53)	0 (0)		2 (12.5)	1 (4.35)
Volunteer	0 (0)	0 (0)	0 (0)		0 (0)	0 (0)		0 (0)	0 (0)
Unemployed	5 (2.59)	2 (2.02)	0 (0)		0 (0)	0 (0)		2 (12.50)	1 (4.35)
Incapacity of work	11 (5.7)	8 (8.08)	0 (0)		1 (1.75)	0 (0)		1 (6.25)	1 (4.35)
Retired	148 (76.68)	73 (73.73)	7 (87.5)		46 (80.70)	8 (100)		11 (68.75)	20 (86.96)
*Education*				0.696			0.067		
Elementary	20 (10.36)	12 (12.12)	1 (12.5)		5 (8.77)	3 (37.50)		1 (6.25)	2 (8.70)
Middle school	28 (14.51)	12 (12.12)	1 (12.5)		8 (14.04)	0 (0)		4 (25)	5 (21.74)
High school	91 (47.15)	44 (44.44)	2 (25)		30 (52.63)	4 (50)		6 (37.5)	22 (47.83)
Higher education	35 (18.13)	22 (22.22)	2 (25)		8 (14.04)	0 (0)		4 (25)	1 (4.35)
University	17 (8.81)	8 (8.08)	2 (25)		5 (8.77)	0 (0)		1 (6.25)	4 (17.39)
PhD	2 (1.04)	1 (1.01)	0 (0)		1 (1.75)	1 (12.50)		0 (0)	0 (0)

AHT: arterial hypertension; BMI: body mass index; DiM: diabetes mellitus; Euroscore II: European System for Cardiac Operative Risk Evaluation.

**Table 2 jcm-11-02674-t002:** Clinical outcomes. Data is represented as *n* (%) or median (IQR).

	TECS (*n* = 193)	Endo-CABG (*n* = 99)	Open CABG (*n* = 8)	Yil-AVR (*n* = 57)	TAVI (*n* = 8)	MVATS (*n* = 16)	Combination (*n* = 23)
CPB time (min)	93.50 (71–177)	78 (58–109)	98.5 (91–118)	94 (79–112)	-	95 ( 86–111)	149 (115–162)
Clamping time (min)	59 (42–75)	50 (32–62.75)	77 (69–98)	62 (55–77)	-	63.50 (49–74)	100 (81–120)
Number of grafts	-	2 (2–3)	3.50 (2.75–4.00)	-	-	-	-
Ventilation time (h)	4 (3–7)	5 (3–7.50)	5.38 (4.88–8.38)	3.25 (2–5)	-	4 (2.13–6.25)	6 (5–22.69)
ICU LOS (h)	42 (23–68)	42.50 (23–67.50)	69.50 (42–78)	27 (23–53)	23.50 (20–24)	24 (22–61)	65.50 (39–141)
Hospital LOS (days)	5 (4–7)	5 (4–6)	6 (4–8)	5 (3–6.25)	3 (2–4.25)	5.50 (4–7.75)	6 (5–10)
Bleeding 24 h (mL)	300 (183.8–600)	335 (187.5–693.8)	700 (557.5–882.5)	235 (160–300)	-	280 (132.5–405)	280 (175–538.8)
Early revision	10 (5.21)	6 (6.06)	0 (0)	2 (3.51)	1 (12.50)	1 (6.67)	1 (4.35)
Late revision	4 (2.08)	1 (1.01)	0 (0)	0 (0)	0 (0)	1 (6.67)	2 (8.70)
*Neurological*							
CVA	2 (1.04)	1 (1.01)	0 (0)	1 (1.75)	0 (0)	0 (0)	0 (0)
TIA	2 (1.04)	2 (2.02)	0 (0)	0 (0)	0 (0)	0 (0)	0 (0)
Epilepsy	1 (0.52)	0 (0)	0 (0)	0 (0)	0 (0)	0 (0)	1 (4.35)
Delirium	1 (0.52)	0 (0)	0 (0)	1 (1.75)	0 (0)	0 (0)	0 (0)
Mortality	8 (4.17)	1 (1.01)	0 (0)	2 (3.51)	0 (0)	0 (0)	5 (21.74)
30-day mortality	5 (2.59)	0 (0)	0 (0)	1 (1.75)	0 (0)	0 (0)	4 (17.39))

CPB: cardiopulmonary bypass; CVA: cerebrovascular accident; ICU: intensive care unit; LOS: length of stay; TIA: transient ischemic attack.

**Table 3 jcm-11-02674-t003:** Results of the univariate and multivariate logistic regression analyses for good QoR at 30 days and 90 days postoperatively.

	Coefficient	SE	OR	95% CI	*p*-Value
A. Univariate factors correlated with 30-day recovery					
Age	−0.040	0.018	0.960	0.928–0.994	0.021
CPB time	−0.009	0.004	0.991	0.983–0.999	0.022
AHT	−0.711	0.341	0.491	0.252–0.958	0.037
Smoking	-	-	-	-	0.048
Euroscore II	0.167	0.085	0.849	0.719–1.002	0.053
ICU LOS	−0.006	0.003	0.994	0.988–1.000	0.052
Hospital LOS	−0.071	0.038	0.931	0.864–1.004	0.062
Education	-	-	-	-	0.075
Combinations	−0.879	0.497	0.415	0.157–1.101	0.077
Clamping time	−0.008	0.005	0.992	0.982–1.001	0.091
B. Multiple regression model for 30-day recovery					
AHT	-0.733	0.349	0.480	0.242–0.952	0.036
ICU LOS	-0.006	0.003	0.994	0.988–1.000	0.046
C. Univariate factors correlated with 90-day improvement					
BMI	0.073	1.064	0.127	0.997–1.159	0.059
Hospital LOS	−0.076	0.042	0.927	0.854–1.006	0.070
Clamping time	0.008	0.005	1.008	0.999–1.018	0.094
D. Multiple regression model for 90-day improvement					
Clamping time	0.011	0.005	1.011	1.001–1.022	0.034
Hospital LOS	−0.095	0.047	0.909	0.829–0.997	0.044

AHT: arterial hypertension; BMI: body mass index; CI: confidence interval; CPB: cardiopulmonary bypass; Euroscore II: European System for Cardiac Operative Risk Evaluation; ICU: intensive care unit; LOS: length of stay, OR: odds ratio, SE: standard error. A *p*-value of <0.05 was considered statistically significant. Good QoR is defined as recovered status at 30 days and improvement at 90 days.

## Data Availability

The data underlying this article will be shared on reasonable request to the corresponding author.

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
