# Peer review of "90-Day Patient-Centered Outcomes after Totally Endoscopic Cardiac Surgery: A Prospective Cohort Study"

_jcm, 2022, doi:10.3390/jcm11092674_

Round 1

Reviewer 1 Report

In this observational cohort study, Claessens J. et al aimed to assess both patient-centered and clinical outcomes after TECS.

HRQL was assessed by the SF-36 and EQ-5D questionnaires. 

The main results were that all types of TECS improved quality of life after 90 days and shorter clamping time and longer hospital LOS were the predictors.

The study is well structured and based on a solid background. Regarding this part, I recommend to put the year (for example of the introduction by Yilmaz et al. of the endo-CABG technique; the other one by Vola et al etc.).

The scientific relevance of the topics is highly appreciated. In spite of this, I believe that it can be improved in the presentation of the methods (it is necessary to divide the statistical part from the rest and give hints on the ethics committee and registration of the study) and of the results (comment on table 1 that has not been done).
Even the discussion section can be improved, dividing it, for example, by subparagraphs.

Regarding the style: when quoting the figures you have to put the space before the number (example Figure 2F and not Figure2F).

Author Response

Response to Reviewer 1 Comments

Dear Mr. Barret Zhang, dear reviewer,

Please find herewith our revised manuscript entitled “90-day patient-centered outcomes after totally endoscopic cardiac surgery: A prospective cohort study”. We would like to thank the reviewer for the constructive comments on our manuscript. We have followed the very helpful suggestions and incorporated a number of improvements based on our response to the reviewers suggestions and comments. These revisions are marked up using the “Track Changes” function.

Point by point response to reviewer comments:

Point 1: The study is well structured and based on a solid background. Regarding this part, I recommend to put the year (for example of the introduction by Yilmaz et al. of the endo-CABG technique; the other one by Vola et al etc.).

Response 1: The years of the introduction of the new techniques were added (line 33 and 39). The year of introduction of the MVATS procedure was already present (line 42).

Point 2: The scientific relevance of the topics is highly appreciated. In spite of this, I believe that it can be improved in the presentation of the methods (it is necessary to divide the statistical part from the rest and give hints on the ethics committee and registration of the study) and of the results (comment on table 1 that has not been done).
Even the discussion section can be improved, dividing it, for example, by subparagraphs.

Response 2: We added subheadings to the method section and added more information regarding the ethical committee and registration of the study (line 60-63). Also we seprarated the statistical part. Concerning the results, a section about the demographics (table 1) was added (line 162-167). We have rewritten some parts of the results section to improve the readability. Additionally, we have added subheading to the discussion and changed the order to have a more logical order of the discussion.

Point 3: Regarding the style: when quoting the figures you have to put the space before the number (example Figure 2F and not Figure2F).

Response 3: We have changed the quoting of the figures.

We hope you are satisfied with our revised manuscript. We think our paper has improved significantly and would like to thank the reviewers for that.

Looking forward to the decision,

Kind regards,

Jade Claessens

Reviewer 2 Report

The aims of this single center, prospective, non- randomized study are to assess both patient-centered and clinical outcomes after minimally invasive surgical   procedures.

Eligible patients undergoing totally endoscopic cardiac surgery (TECS), TAVI, or conventional open CABG between November 2019 and October 2020 were enrolled. The primary outcome of this study is HRQL after different types of TECS procedures. HRQL is assessed by the SF-36 and EQ-5D questionnaires.

Two- hundred nine patients were enrolled, of these, 193 patients underwent totally endoscopic cardiac surgery, eight underwent cardiac surgery through a sternotomy, and eight underwent transcatheter aortic valve implantation. Quality of life was determined through the Short Form 36 and European quality of life 5-dimension questionnaires before the surgery and 14, 30, and 90 days afterward. In patients who underwent totally endoscopic cardiac surgery, the quality of life improved over the three time periods. The different domains of the questionnaire evolved in a positive manner. However, 14 days postoperatively, a decline in quality of life was noted, followed by a return to baseline at 30 days and an increase at 90 days. Authors concluded that totally endoscopic cardiac surgery improves the quality of life 90 days after surgery.

Comments/ Suggestions:

Page 2: Under methods:

1st paragraph:

>>>>Please briefly describe endo-CABG technique.

5th paragraph: “: QoR was predefined into recovered and improved. Recovered was defined as the absence of a significant difference between the total median postoperative SF-36 score and baseline SF-36 score. Improved was described as a substantial improvement in total median postoperative SF-36 score compared with baseline.”

>>>>>QoR was predefined into recovered and improved. Choosing the word “recovered “is misleading. Per this definition another group name should be used such as “failed to improve”

“An SF-36 score between one point lower and four points higher 88 than the baseline SF-36 score was considered recovered. An SF-36 score five points higher than base- 89 line was considered improved.”

>>>>> Please explain how these criteria are created. Based on what??

Page 3:

Table 1.

>>> Please include p values

Author Response

Response to Reviewer 2 Comments

Dear Mr. Barret Zhang, dear reviewer,

Please find herewith our revised manuscript entitled “90-day patient-centered outcomes after totally endoscopic cardiac surgery: A prospective cohort study”. We would like to thank the reviewer for the constructive comments on our manuscript. We have followed the very helpful suggestions and incorporated a number of improvements based on our response to the reviewers suggestions and comments. These revisions are marked up using the “Track Changes” function.

Point by point response to reviewer comments:

Point 1: Please briefly describe endo-CABG technique.

Response 1: We added the description of the endo-CABG, Yil-AVR and MVATS procedures (line 76-83).

Point 2: QoR was predefined into recovered and improved. Choosing the word “recovered “is misleading. Per this definition another group name should be used such as “failed to improve”

Response 2: The patients who are recoverd return to baseline levels after the surgery. When patients reach better levels compared to baseline, these patients are improved. As indicated on line 143 and 144, there are other group names that indicate that these patients are not recovered or not improved. We have changed these names to “failed to recover” and failed to improve. These defenitions are based on two other manuscripts that also investigated the recovery of patients. In Stessel et al. (2021) [1], they use the following defenitions:

“Recovered was defined as the absence of a significant difference between total median postoperative FRI score and baseline FRI score. Improved was defined as a significant improvement of total median postoperative FRI score compared with baseline”

The other manuscript of Bratwall et al.(2010)[2], used these definitions:

Recovery was predefined into improved and recovered in order to achieve a better resolution of the recovery process, having in mind that the procedures studied may or may not be related to pre-operative symptoms.

Improved defined as the time when the proportion of patients with symptoms was lower than that at base line.

Recovered defined as the time when the mean number of surgery-related symptoms (pain, immobilisation, depression and sleep disturbances) was <0.5.

We added the references to the manuscript.

Point 3: Please explain how these criteria are created. Based on what??

Response 3: These criteria were based on two previously published manuscripts [1, 3]. These both also assesed the predictors of recovery, however, with other tools to asses the recovery. We defined these criteria in a similar way. The references were added in the manuscript.

Point 4: Please include p values in Table 1

Response 4: P-values are added in table 1. We added them to compare Endo-CABG and open CABG; and to compare Yil-AVR with TAVI. For the other sub-types of operations we do not make a comparision because we do not find it relevant since we do not compare the results of these subpopulations.

  1. Stessel, B., et al., One-month recovery profile and prevalence and predictors of quality of recovery after painful day case surgery: Secondary analysis of a randomized controlled trial. PLoS One, 2021. 16(1): p. e0245774.
  2. BRATTWALL, M., et al., Patients' assessment of 4-week recovery after ambulatory surgery. Acta Anaesthesiologica Scandinavica, 2011. 55(1): p. 92-98.
  3. Stessel, B., et al., Prevalence and Predictors of Quality of Recovery at Home After Day Surgery. Medicine, 2015. 94(39): p. e1553-e1553.

We hope you are satisfied with our revised manuscript. We think our paper has improved significantly and would like to thank the reviewers for that.

Looking forward to the decision,

Kind regards,

Jade Claessens

Round 2

Reviewer 2 Report

Authors have incorporated most of my suggestions and revised the MS accordingly.